# Observed joint infection incidence following needle arthroscopy performed in operating and nonoperating room environments in client-owned dogs: A retrospective cohort study

**Alessandra Chiaramonte** ⬥*, **Cassio R. A. Ferrigno, Darryl L. Millis, Jessica W. Montoya**

Department of Surgery, University of Tennessee College of Veterinary Medicine (UT-CVM), Knoxville, Tennessee, United States of America

* achiaram@utk.edu

## Abstract

### Objective

To report the use of needle arthroscopy (NA) on an outpatient basis and to compare joint infection rates following needle arthroscopy performed under general anesthesia in an operating room (OR) versus heavy sedation in a nonoperating room (NOR) clinical environment in client-owned dogs. We hypothesized that there would be no observed increase in infection risk dependent on location in this cohort.

### Study design

A retrospective study of 75 dogs, inclusive of 90 individual joints, that underwent needle arthroscopy at one academic institution from May 4th, 2022, through December 20th, 2024. A consent form authorizing the use of medical information for clinical research was obtained for every dog undergoing treatment upon admission to the hospital.

### Results

After excluding for pre-existing infection, fifty-six joint NA (45 dogs) were performed in an operating room under general anesthesia, while thirty-four joint NA (30 dogs) were performed under heavy sedation in a nonoperating room environment. Routine follow-up was collected either in clinic or by referring veterinarian examinations for NA OR (57.8 ± 0.87 days, range 7–365 days) and NA NOR (42.6 ± 1 day, range 7–425 days). One case in the OR group was lost to follow-up. No inferential statistical analysis was performed due to no observed infections in either treatment group.

**Data availability statement:** All relevant data are within the manuscript.

**Funding:** The author(s) received no specific funding for this work.

**Competing interests:** The authors have declared that no competing interests exist.

## Conclusion

In this cohort, no postoperative joint infections were observed following outpatient needle arthroscopy performed under heavy sedation when appropriate aseptic technique was used.

## Introduction

Septic arthritis following joint arthroscopy is a rare but deleterious postoperative complication. The reported infection rate for conventional arthroscopy in small animal orthopedics ranges from 1–3% [1,2]. A retrospective study following 353 elective conventional stifle arthroscopies, diagnosed only 3 cases with postoperative septic arthritis (0.85%) [2]. A study on canine elbow conventional arthroscopy revealed a lower incidence of postoperative septic arthritis at 0.22% [3]. Although several articles have reported infection rates for conventional arthroscopy, there is a paucity of data on infection rates with needle arthroscopy.

Needle arthroscopy (NA) has increased in popularity in the veterinary clinical setting over the last five years [4–7]. This is due to the minimally invasive diagnostic accuracy of needle arthroscopy with less equipment and decreased preparation time [7]. Several articles have been published validating the diagnostic accuracy of needle arthroscopy over conventional arthroscopy for medial coronoid disease, medial shoulder instability, and medial meniscal tears in dogs [4–7]. Needle arthroscopy also has reported utility in treating medial coronoid disease via fragment retrieval with favorable outcomes [8]. Needle arthroscopy offers the ability to perform minimally invasive diagnostics on an outpatient basis with less equipment. Sedated arthroscopy avoids undergoing general anesthesia which can spare considerable costs, accessibility, and time to diagnosis. However, it remains unclear whether outpatient sedated arthroscopy poses a greater risk of infection due to differences in environmental sterility and ventilation compared to those in a standard operating room.

Despite the increase in usage of needle arthroscopy, there have been no published studies assessing the incidence of infection when used on an outpatient basis outside of an operating room in dog joints. The objective of this study is to compare postoperative joint infection rates following needle arthroscopy performed in a sterile operating room (OR) versus heavy sedation in a nonoperating room (NOR) clinical environment with appropriate aseptic technique in client-owned dogs. We hypothesized that there would be no observed increase in infection risk dependent on location in this cohort.

## Materials and methods

Medical records of all dogs undergoing NA at University of Tennessee College of Veterinary Medicine (UT-CVM) were compiled from May 4th, 2022, through December 20th, 2024. Prospective animal research ethics committee approval was not required as this was a retrospective clinical study utilizing existing

medical records. Formal consent to terms of conditions and treatment was signed by each client for surgical treatment. The formal consent form can be found in the supplemental documents. This formal consent authorizes UT-CVM to use medical records for retrospective research purposes. No animals were sacrificed or euthanized for this study.

Researchers had access to identification of patients involved in this study and were not blinded. Clinical records were obtained for a total of 85 dogs undergoing NA for 100 joints. Ten joints in ten dogs were excluded from the study due to previous confirmed evidence of infection prior to needle arthroscopy (nine OR, one NOR case). Additional data attained from the medical records included signalment, age at the time of surgery, gender, breed, dog size, laterality of clinical signs, and any pre-existing conditions that may affect infection rates such as endocrinopathies, obesity, and concurrent infections. Dog size was stratified according to weight. Small dogs were less than 10 kg, medium dogs weighed 11–20 kg, and large dogs weighed greater than 21 kg. Intra-operative complications, including conversion to an open procedure, need for a second arthroscopic procedure, or worsening lameness/ unresolved lameness because of osteoarthritis, were collected. Postoperative complications such as septic arthritis, incision dehiscence, or incision infection were compiled. Clinical suspicion of septic arthritis or joint associated infection included evidence of joint effusion with arthrocentesis analysis of joint fluid suggestive of infection based on high cellular appearance, predominant neutrophil population of greater than 40 percent neutrophils, or a positive bacterial culture from the affected joint [1].

All NA procedures were performed with a sterilized 1.9 mm Arthrex Nanoscope Camera and Visualization System (Arthrex Vet Systems, Naples, Florida, United States) (Fig 1). The location for the needle arthroscopy procedure, OR under general anesthesia versus outpatient NOR under heavy sedation, was determined based on the diagnosis. If a patient required surgical intervention along with arthroscopy, both procedures were performed in the OR. If the physical and orthopedic examination did not indicate surgical intervention or surgical intervention was to be performed later, NA was performed on an outpatient basis (NOR) (Fig 2). The NOR arthroscopic procedures were performed in a non-sterile daily treatment room or physical therapy modalities room. Neither of these locations were controlled for foot traffic or ventilation during the time of the procedure.

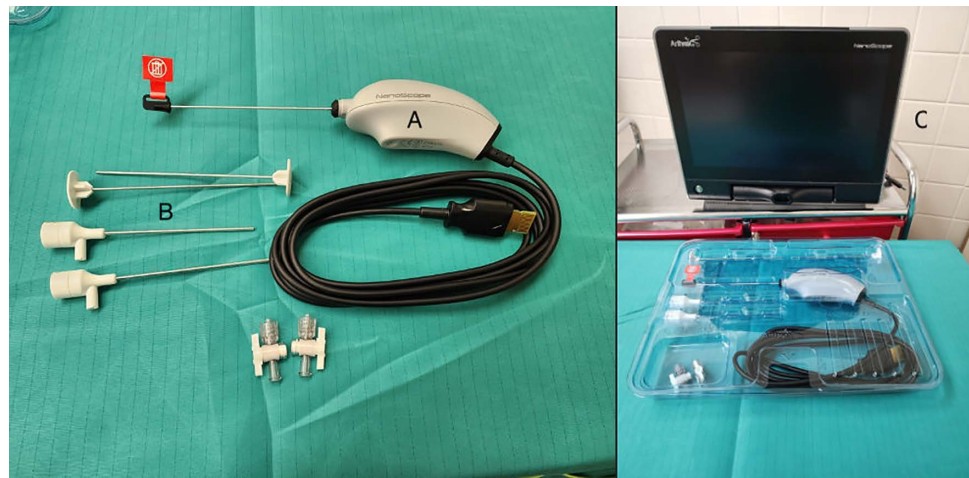

**Fig 1. Needle arthroscopy equipment and visualization system.** Components of the 1.9 mm Arthrex needle arthroscopy system used in this study, including the needle arthroscopy (A), disposable cannulas and obturators (B), and portable visualization monitor (C). The 1.9 mm needle arthroscopy and monitor allow for ease of image acquisition and diagnosis without a conventional arthroscopy tower. The limited equipment also facilitates use in nonoperating room environments.

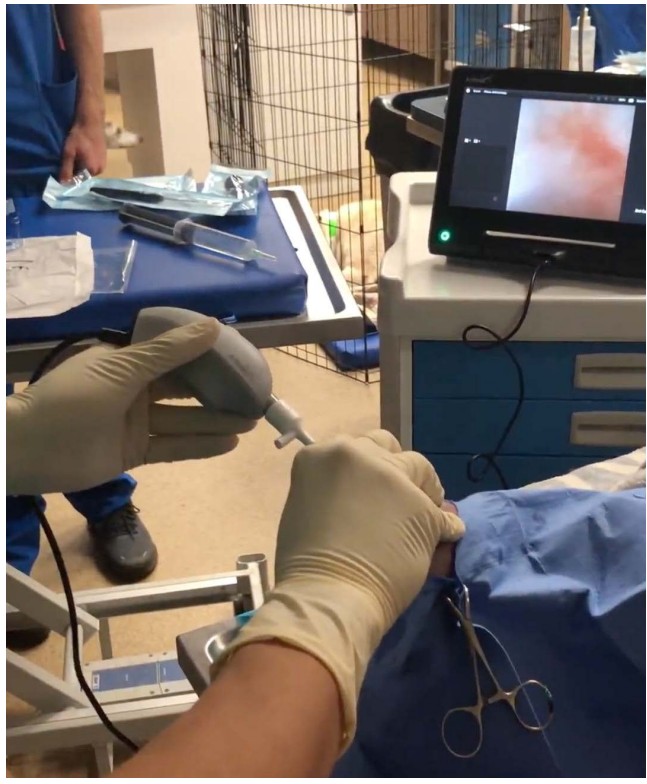

**Fig 2. Outpatient needle arthroscopy performed in a nonoperating room environment.** Demonstration of a procedural setup for outpatient needle arthroscopy performed under heavy sedation outside of a standard operating room. The environment is not limited by foot traffic as it is performed in a treatment room. The needle arthroscope is introduced percutaneously into the stifle joint, with visualization displayed on the portable monitor next to the patient.

All surgical procedures were performed by faculty surgeons or a surgical resident under direct supervision. An anesthetic protocol was tailored to each individual patient based on anesthesiologist or clinician preference. Patients undergoing an additional procedure were prepared with a standard protocol of 4% chlorhexidine scrub (BD E-Z Scrub impregnated with 3% chloroxylenol, Becton Dickinson and Company, Franklin Lakes, New Jersey, United States) followed by a dilute 2% chlorhexidine scrub (Chloradine Scrub 2%, Aspen Veterinary Resources, Liberty, Missouri, United States) after clipping the area. The surgical site was then sterilely prepared with ChloraPrep (2% chlorhexidine gluconate & 70% Isopropyl alcohol, Becton Dickinson and Company, Franklin Lakes, New Jersey, United States) for 3 minutes prior to initial skin incision in the operating room. Patients undergoing NA in the OR also received a prophylactic dose of cefazolin (22 mg/kg, 100 mg/mL, 1 gm/vial, NovaPlus, Hikma Pharmaceuticals, Berkeley Heights, New Jersey, United States) intravenously in preparation for proposed surgical procedures and prolonged anesthesia times (ex. TPLO).

An alternating dilute 2% chlorhexidine and 70% alcohol scrub was performed on joints undergoing NA outside of the operating room (NOR). The aseptic solution was allowed to dry prior to the initial skin incision. The environment for NOR was either the Orthopedic Service treatment area or Physical Therapy modalities room. In either of these areas, foot traffic was not restricted and ventilation was not controlled. Outpatient NOR NA patients did not receive prophylactic antibiotics as antibiotics are not indicated for short, clean procedures not involving implants and can lead to an increased risk of antimicrobial resistance. However, this is still a controversial topic even in human medical literature [9–12]. The joint was then covered with a single use sterile Sklar Clear Utility Drape (Sklar Surgical Instruments, Sklar Corporation, West Chester, Pennsylvania, United States).

## Exclusion criteria

Ten cases exhibited symptoms concerning infection resulting from a previous procedure; thus, they were excluded from the data set. Of these ten cases, eight had evidence of infection prior to NA and were scheduled for a TPLO implant removal. These cases were explanted with positive culture growth of implant screws. One case was infected due to a chronic bite wound. The last case was excluded for a canine unicompartmental elbow (CUE) implant associated infection.

## Follow-up

Follow-up was determined by the case, according to the diagnosis and procedure. There was no standard recheck protocol. All cases were followed for a minimum of 7–14 days, exceeding the typical postoperative detection of septic arthritis. The follow-up duration in OR cases was longer due to concurrent surgical procedures requiring scheduled postoperative rechecks. Recheck incision evaluations were performed by the primary care veterinarian, Orthopedic Surgery Service, or Physical Therapy and Sports Medicine Service within 2 weeks. Most commonly, recheck radiographs were performed at eight weeks following tibial plateau leveling osteotomy (TPLO). Only one of the OR cases was lost to follow-up.

## Statistical analysis

Descriptive statistics such as signalment were reported. The mean, median, standard deviation, and range were calculated for age, weight, and follow-up period. No infections were observed for either NA in the OR and NOR, precluding inferential statistical analysis.

## Results

NA was performed in 75 cases, including 90 individual joints. Fifty-six needle arthroscopies in forty-five dogs were performed under general anesthesia in an OR, while thirty-four needle arthroscopies in thirty dogs were performed under heavy sedation in NOR on an outpatient basis (Tables 1–2).

The mean age of patients at the time of the procedure was 5.6 years for OR and 5.9 years for NOR. OR cases were as follows: eight dogs intact males, fourteen were castrated males, three intact females, and twenty spayed females. NOR cases were eight intact males, ten castrated males, two intact females, and ten spayed females. Fifty-five dogs were categorized as large (≥ 21 kg), fourteen medium (11 kg – 20 kg), and six small (> 10 kg). The average weight was 27.4 kg in OR and 27.5 kg in NOR. Large breed dogs were the most represented group in this study population. It is reasonable to postulate that this is due to higher incidence of orthopedic conditions able to be treated in a minimally invasive manner in this population [3–8]. Five cases had repeat arthroscopies for the same previously diagnosed issue: one OR, two NOR, and two initially performed NOR then moved into the OR based on diagnosis. The two cases moved into the operating room underwent general anesthesia and thus were included in the OR group. Second look arthroscopies were performed in these cases for persistent or progressive lameness due to meniscal tear or progression of osteoarthritis. One second look arthroscopy case was concerning for potential infection, but repeat cultures performed on synovial fluid showed no evidence of bacterial growth. Ten patients had more than one joint arthroscopically examined under the same anesthetic or sedative event (7 OR, 3 NOR cases). These joints were examined individually.

A total of 90 individual joints underwent NA including one right carpus, four left cubital joints, five right cubital joints, seven left glenohumeral joints, twelve right glenohumeral joints, one left tarsal, twenty-four left stifles, twenty-seven right stifles, five left coxofemoral, and four right coxofemoral joints (Tables 1–2). Diagnoses included medial coronoid disease, biceps/supraspinatus tendinopathy, complete or partial cranial cruciate ligament ruptures, medial meniscal tears, patellar tendon desmitis, or coxofemoral osteoarthritis. Nine patients underwent a second-look stifle arthroscopy after a TPLO to assess for possible late meniscal tear or other postoperative complications. Intraoperative complications were recorded, including conversion to an open tarsal and stifle arthrotomy procedure due to difficulty with fragment removal in two cases. No postoperative complications were associated with the needle arthroscopy.

**Table 1. Summary of needle arthroscopy cases performed in the operating room (OR).**

| OR NA Cases | 56 individual joints (45 dogs) | |
|---|---|---|
| Average Age | 5.6 years | |
| Average Weight | 27.4 kg | |
| Follow-Up | Mean 57.8 (± 0.87) days<br>Range 7–365 days<br>1 case lost to follow-up | |
| Sex | | |
| Intact Male | 8 | |
| Castrated Male | 14 | |
| Intact Female | 3 | |
| Spayed Female | 20 | |
| Joint | Number of Joints per Group | Diagnosis |
| Carpus | Left = 0 | Incomplete ossification of radial carpal bone |
| | Right = 1 | |
| Cubital | Left = 3 | Fragmented medial coronoid (FMCPs), elbow dysplasia |
| | Right = 4 | |
| Glenohumeral | Left = 4 | Medial glenohumeral ligament tear, supraspinatus tendon tear, biceps tenosynovitis |
| | Right = 6 | |
| Tarsal | Left = 1 | Osteochondritis dissecans (OCD) |
| | Right = 0 | |
| Stifle | Left = 19 | Cranial cruciate ligament rupture (CCLR), medial meniscal tear, osteoarthritis |
| | Right = 16 | |
| Coxofemoral | Left = 2 | Hip dysplasia, osteoarthritis |
| | Right = 0 | |

Out of the total ninety joints undergoing NA, there was no observed evidence of infection in either group despite less stringent controls on the outpatient NOR NA (56 OR, 34 NOR). Since zero infectious events were observed, inferential statistical comparison of infection rates could not be performed. Therefore, the upper 95% confidence limits for infection risk were estimated using the rule of three, a binomial-based approximation for zero-event results. Using this method, the upper confidence limit for infection risk was 5.4% in the OR cases and 8.8% in the NOR cases. Based on these estimates, the true risk of infection is unlikely to exceed 5% in the operating room group and 8% in the outpatient group. This acknowledges that there is uncertainty due to small sample size.

The mean time for follow-up of cases was 57.8 (± 0.87) days for OR (range 7–365 days) and 42.6 (± 1.00) days for NOR (range 7–425 days) post-arthroscopy. During this time, no cases were noted to have concerns of septic arthritis based on physical examination, orthopedic examination, and/or radiographic evaluation. Only one case in the OR group was lost to follow-up. Follow-up was variable dependent on diagnosis. All cases in either OR or NOR groups were evaluated by either a primary care veterinarian or Orthopedic Surgery Service to confirm healed incision sites. Most cases in the OR group then underwent follow-up radiographs with Orthopedic Surgery Service in 8 weeks for TPLOs. Cases in the NOR groups were scheduled for follow-up surgeries shortly after their diagnosis. Additional rechecks were scheduled with the Orthopedics Service only if lameness recurred.

## Discussion

In this retrospective cohort of client-owned dogs, no postoperative infections were observed following NA performed in the OR or outpatient NOR. Although the absence of infectious events precluded inferential statistical comparisons, these

**Table 2. Summary of needle arthroscopy cases performed outpatient in a nonoperating room clinical environment (NOR).**

| NOR NA Cases | 34 individual joints (30 dogs) | |
|---|---|---|
| Average Age | 5.9 years | |
| Average Weight | 27.5 kg | |
| Follow-Up | Mean 42.6 (± 1.00) days<br>Range 7–425 days<br>0 cases lost to follow-up | |
| Sex | | |
| Intact Male | 8 | |
| Castrated Male | 10 | |
| Intact Female | 2 | |
| Spayed Female | 10 | |
| Joint | Number of Joints per Group | Diagnosis |
| Carpal | Left = 0 | |
| | Right = 0 | |
| Cubital | Left = 1 | Fragmented medial coronoid (FMCPs), elbow dysplasia |
| | Right = 1 | |
| Glenohumeral | Left = 3 | Osteoarthritis, cartilage erosion of humeral head, supraspinatus tendon tear, biceps tenosynovitis |
| | Right = 6 | |
| Tarsal | Left = 0 | |
| | Right = 0 | |
| Stifle | Left = 5 | Cranial cruciate ligament rupture (CCLR), medial meniscal tear, osteoarthritis, patellar desmitis |
| | Right = 11 | |
| Coxofemoral | Left = 3<br>Right = 4 | Osteoarthritis, hip dysplasia |

findings do suggest that, within the parameters of this study, outpatient NA performed outside a standard operating room environment was not associated with an observed increase in postoperative joint infection risk when appropriate aseptic technique was used.

In human medical and veterinary patients, septic joint arthritis is uncommon, but can result in detrimental effects including cartilage damage, permanent loss of limb function, or lead to limb amputation when not addressed promptly and aggressively [1,13]. Reported infection rates following conventional arthroscopy in dogs ranges from 0.2–3% [1–3]. While infection risk has been described for conventional arthroscopy, the comparable data for NA – particularly when performed outside of a controlled operating room environment – is lacking. To the authors knowledge, this is the first study to document infection rates for outpatient NA performed outside of an operating room in veterinary patients. The results of this cohort suggest that NA may represent a feasible cage side diagnostic test on an outpatient basis in sedated patients without an observed increased risk of infection even under less stringent environmental controls, including absence of controlled ventilation, unrestricted foot traffic, and the omission of perioperative antimicrobial prophylaxis. Additionally, the feasibility of NA on an outpatient basis provides several advantages. These include reduced anesthetic risk, decreased procedural cost, improved accessibility, less equipment, and shortened time to diagnosis.

Several limitations should be considered when interpreting the results of this paper. A major limitation to this study is the retrospective nature and relatively small sample size. This may raise the possibility of type II error. Ideally, a total of 514 needle arthroscopies (257 in the operating room and 257 outpatient) would be required to achieve a strong power study of 0.8 and alpha of 0.05 (power study performed using GraphPad Prism 9.4.1, Dotmatics, 2025). Additionally, we recognize that bacterial cultures were not performed routinely with each case. Cultures were only performed if there was

concern or suspicion for septic joint effusion based on clinical signs, radiography, and cytology. Finally, differences in anesthetic protocols and not routine follow-up between groups can introduce confounding factors that could not be controlled within this study design.

Future investigations should include a prospective, multi-institutional study with standardized preoperative and postoperative synovial fluid analysis and follow-up protocols. Routine, standardized pre and post NA synovial fluid analysis and cultures would be more accurate at defining infection risk associated with needle arthroscopy in different clinical environments.

In conclusion, within the limitations of this retrospective cohort, outpatient needle arthroscopy was not associated with observed increase in joint infections when appropriate aseptic technique was used. The ability to perform needle arthroscopy safely in an outpatient clinical environment may improve diagnostic accessibility while reducing anesthetic concerns and cost for canine patients. These findings support continued exploration in minimally invasive needle arthroscopy on an outpatient diagnostic modality veterinary orthopedic field.

## Author contributions

**Conceptualization:** Alessandra Chiaramonte, Cassio R. A. Ferrigno.

**Data curation:** Alessandra Chiaramonte, Darryl L. Millis, Jessica W. Montoya.

**Investigation:** Alessandra Chiaramonte, Cassio R. A. Ferrigno.

**Methodology:** Alessandra Chiaramonte, Cassio R. A. Ferrigno.

**Resources:** Alessandra Chiaramonte, Cassio R. A. Ferrigno.

**Visualization:** Cassio R. A. Ferrigno.

**Writing – original draft:** Alessandra Chiaramonte.

**Writing – review & editing:** Alessandra Chiaramonte, Cassio R. A. Ferrigno, Darryl L. Millis, Jessica W. Montoya.

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
