## [Decision Letter · Decision Letter 0]

10 Nov 2025

Dear Dr. Chiaramonte,

Thank you for submitting your manuscript to PLOS ONE. After careful consideration, we feel that it has merit but does not fully meet PLOS ONE’s publication criteria as it currently stands. Therefore, we invite you to submit a revised version of the manuscript that addresses the points raised during the review process.

We look forward to receiving your revised manuscript.

Kind regards,

Xiaoen Wei

Academic Editor

PLOS ONE

Journal Requirements:

2. Please include a separate caption for each figure in your manuscript.

Reviewers' comments:

Reviewer's Responses to Questions

**Comments to the Author**

1. Is the manuscript technically sound, and do the data support the conclusions?

Reviewer #1: Yes

Reviewer #2: Partly

2. Has the statistical analysis been performed appropriately and rigorously?

Reviewer #1: Yes

Reviewer #2: N/A

3. Have the authors made all data underlying the findings in their manuscript fully available?

Reviewer #1: Yes

Reviewer #2: Yes

4. Is the manuscript presented in an intelligible fashion and written in standard English?

Reviewer #1: Yes

Reviewer #2: Yes

Reviewer #1: Dear Author,

You can find my comments below,

Best regards

Title

• Consider simplifying wording for clarity:

“Comparison of Infection Rates for Needle Arthroscopy in Operating Room vs. Outpatient Sedated Settings in Dogs”

• Avoid overly long phrasing; keep terminology consistent throughout the manuscript.

Abstract

• State follow-up duration more explicitly (mean ± range).

• Add a sentence clarifying no inferential statistics were performed due to zero infection event rate.

• Mention the study design as retrospective in the first or second sentence.

Introduction

• Clarify why infection rates may theoretically differ between OR and outpatient sedated settings (e.g., environmental sterility differences).

• Strengthen motivation: emphasize the clinical relevance of avoiding general anesthesia and using minimally invasive diagnostic options.

Methods

1. Infection definition must be clearly stated.

Define what constitutes septic arthritis (e.g., clinical signs, cytology, culture positivity, response to treatment).

2. Outpatient procedural environment should be described more thoroughly:

• Room type

• Draping and barrier techniques

• Number of personnel

• Air handling/ventilation, if relevant

3. Follow-up description is insufficient:

• Specify how follow-up was conducted (in-person exam vs. phone follow-up vs. primary veterinarian exam).

• Indicate how many cases were lost to follow-up and how these were handled analytically.

4. Antibiotic protocol rationale should be briefly justified (why no prophylaxis in NOR).

Results

• Present follow-up duration for each group clearly and consistently.

• Specify how many cases had repeat arthroscopy and whether this was related to unresolved pathology vs. potential infection concerns.

• If possible, include a small table summarizing follow-up type (clinic vs. referring vet vs. phone).

Discussion

• Expand on the clinical implications of being able to safely perform NA outside the OR (cost, anesthesia avoidance, accessibility).

• Limitations should be more prominently discussed:

• Retrospective design

• Small sample size

• Zero-event data preventing statistical comparison

• Variable and sometimes short follow-up

• Lack of routine post-procedure cultures

• Consider rephrasing the conclusion to avoid overgeneralization:

The data suggest that needle arthroscopy performed under outpatient sedation did not demonstrate increased infection risk in this cohort when appropriate aseptic technique was used.

Conclusion

• Avoid implying equivalence without statistical testing.

• Use cautious phrasing: “no observed difference” rather than “no difference.”

Reviewer #2: The authors can be commended for bringing this information forward to share. That said I have concerns with the thrust of the paper being a determination of an infection rate without any infections and a too small of a population to be confident that the lack of infections isn't a type 2 error. Their cited sources regarding infections rates for arthroscopy all point to needing a larger population that that presented - unless you combine both groups which would not be recommended as they have significantly different antisepsis protocols which could also effect infection rate. I have attached more in-depth comments in a PDF of their manuscript.

**Do you want your identity to be public for this peer review?** For information about this choice, including consent withdrawal, please see our Privacy Policy

Reviewer #1: **Yes:** Mustafa Akkaya

Reviewer #2: No

---

## [Author Response · Author response to Decision Letter 1]

14 Jan 2026

Dear Plos One Reviewers,

Thank you for your thorough review and comments of our manuscript titled “A comparative analysis of the infection rates associated with needle arthroscopy conducted in an operating room versus those performed under sedation in a nonoperating room setting in canine joints: a retrospective study” retitled Observed Joint Infection Incidence Following Needle Arthroscopy Performed in Operating and Nonoperating Room Environments in Client-Owned Dogs: A Retrospective Cohort Study. The authors appreciate your valuable insight which has allowed us to improve the quality of our original manuscript. We have addressed the specific comments raised in detail below.

Reviewer #1: Mustafa Akkaya

1. Comment: Title consider simplifying wording and avoid long phrasing

a. Response: We agree with your suggestion. The title of our manuscript has been revised and shortened to more explicitly relay the purpose and results of this research.

2. Comment: Abstract. State follow-up duration more explicitly. Ass a sentence clarifying no inferential statistics were performed due to zero infection event rate. Mention study design as a retrospective.

a. Response: These have all been addressed in the revised manuscript and clearly stated.

3. Comment: Introduction. Clarify why infection rates may theoretically differ between OR and outpatient sedated settings. Strengthen motivation: emphasize the clinical relevance of avoiding general anesthesia and using minimally invasive diagnostic options.

a. Response: This has been further clarified and addressed in lines 61-66 in the revised manuscript.

4. Comment: Methods.

a. Clearly state infection definition.

i. Response: Definition of septic arthritis or joint associated infection included lines 95-98

b. Thoroughly describe outpatient procedural environment.

i. Response: Lines 105-108

c. Specify how follow-up was conducted. Indicate how many cases were lost to follow-up.

i. Response: Lines 139-147

d. Rationale for antibiotic protocol should be briefly justified.

i. Response: Please see explanation for rationale in lines 118-131. Included are four human references that this continues to be a controversial topic.

5. Comment: Results. Clarify follow-up duration. Specify how many cases had repeat arthroscopy and whether this was related to unresolved pathology versus potential infection concerns. If possible include a small table summarizing follow-up type.

a. Response: Duration has been clarified in the tables provided. Follow-up was also more clearly explained in lines 1390147. Further clarification was included for the repeat arthroscopy procedures lines 174-177. Only one arthroscopic case was concerning for infection and repeated cultures of synovial fluid showed no growth.

6. Comment: Discussion. Expand on clinical implications of being able to safely perform NA outside the OR. Limitations should be more prominently discussed.

a. Response: We have included an expansion of why it is important to be able to perform NA outside the OR due to decreased cost to client without general anesthesia, bench side diagnostic test that offers quick results, and limits invasive techniques. We outright acknowledge the possibility of type II error and clarify that the absence of observed infections does not equate to proof of zero risk.

7. Comment: Conclusion. Avoid implying equivalence without statistical testing. Use cautious phrasing.

a. Response: We agree with your comments. Our conclusion emphasizes that no infections were observed rather than implying a definitive absence of infection risk. It now clearly states this study as preliminary research that justifies a larger prospective clinical trial in the future.

Reviewer #2:

Dear Reviewer #2,

Thank you for your evaluation of our manuscript and insightful critiques. We appreciate your points regarding the small study population and your concern about meeting our objective. We agree with the reviewer that provided with the low expected incidence of post-arthroscopic infections and limited sample size, our study is not powered to determine true infection rate or detect rare adverse events. We have edited this distinction in our manuscript accordingly by emphasizing the observational assessment of post-arthroscopic infections and changed the language throughout the manuscript to avoid implying definitive infection rate was established. We did change our objective to be two-fold – one, to report the use of outpatient needle arthroscopy since it has not been published in veterinary medicine as an outpatient modality, and to compare our observed infections between nonoperating and operating room environment.

Materials & Methods: Per your recommendation, the “Conditions of Admission/Treatment” aka consent form was moved to supplemental documentation.

Line 172-173: Further addressed that the two cases that were moved into the OR underwent general anesthesia for a longer surgical procedure and thus were included in the OR group.

Line 177-179: If a patient had multiple joints scoped, they were counted individually. Yes, this would have been interesting if any of the joints became infected!

To your point on antibiotics – Our other reviewer also expressed these concerns. Please refer to lines 118-131.

Discussion: We added a dedicated paragraph to our discussion section that addresses expected infection incidence based on published literature and explains why our sample size limits detection of rare events. The power analysis we felt conceptualized the estimate for future prospective work. We outright acknowledge the possibility of type II error and clarify that the absence of observed infections does not equate to proof of zero risk.

Conclusion: We agree and now emphasize that no infections were observed rather than implying a definitive absence of infection risk. The concluding paragraph now clearly states this study as preliminary research that supports feasibility and justifies a larger prospective clinical trial.

We are grateful for the insightful critiques, which have substantially improved the clarity, accuracy, and scientific rigor of our manuscript. We believe that this revised version now appropriately displays our findings, acknowledges our limitations, and contributes meaningful preliminary data regarding outpatient needle arthroscopy in canines.

We hope that these revisions meet your expectations and enhance the clarity of our research. Thank you once again for your constructive feedback and time.

Sincerely,

Alessandra Chiaramonte, VMD

University of Tennessee College of Veterinary Medicine

Small Animal Surgery Department

achiaram@utk.edu

---

## [Decision Letter · Decision Letter 1]

13 Feb 2026

Observed Joint Infection Incidence Following Needle Arthroscopy Performed in Operating and Nonoperating Room Environments in Client-Owned Dogs: A Retrospective Cohort Study.

PONE-D-25-49231R1

Dear Dr. Chiaramonte,

We’re pleased to inform you that your manuscript has been judged scientifically suitable for publication and will be formally accepted for publication once it meets all outstanding technical requirements.

Kind regards,

Xiaoen Wei

Academic Editor

PLOS One

Additional Editor Comments (optional):

Reviewers' comments:

Reviewer's Responses to Questions

**Comments to the Author**

Reviewer #2: All comments have been addressed

Reviewer #3: (No Response)

2. Is the manuscript technically sound, and do the data support the conclusions?

Reviewer #2: Yes

Reviewer #3: Yes

3. Has the statistical analysis been performed appropriately and rigorously?

Reviewer #2: N/A

Reviewer #3: N/A

4. Have the authors made all data underlying the findings in their manuscript fully available?

Reviewer #2: Yes

Reviewer #3: Yes

5. Is the manuscript presented in an intelligible fashion and written in standard English?

Reviewer #2: Yes

Reviewer #3: Yes

Reviewer #2: The authors have addressed my concerns to the best of their ability and as the data and study allows. I have no further suggestions or recommendations. Thank you for giving me the option to review the manuscript revision.

Reviewer #3: This retrospective study by Chiaramonte et al. reports that needle arthroscopy procedures performed in non-operating room environments do not have a higher incidence of post-operative infections when compared to those performed in operating rooms in the population cohort studied. In the first round of review, the reviewers raised concerns regarding the low sample number and requested clarifications regarding some of the methodology and results reported.

In response, the authors have thoroughly revised the manuscript, added a detailed discussion on the limitations of the small cohort size and provided clarifications about the methods and results wherever requested. The manuscript in its current form is acceptable.

I have a few minor suggestions that the authors may add for better readability and interpretability of the manuscript:

1.For summary of cases, the authors may consider combining the two tables and adding separate columns for OR and NOR, for easier comparison between the cases across the groups.

2.The authors mention that repeat culture was performed only for one case and note as a limitation that cultures were not performed for all cases. The authors could clarify in how many cases culture was performed overall and provide the data regarding the same. They could also provide the data regarding the percentage of neutrophil population wherever available.

**Do you want your identity to be public for this peer review?** For information about this choice, including consent withdrawal, please see our Privacy Policy

Reviewer #2: No

Reviewer #3: **Yes:** Reema Banarjee

---

## [Editor Report · Acceptance letter]

PONE-D-25-49231R1

PLOS One

Dear Dr. Chiaramonte,

I'm pleased to inform you that your manuscript has been deemed suitable for publication in PLOS One. Congratulations! Your manuscript is now being handed over to our production team.

Kind regards,

on behalf of

Dr. Xiaoen Wei

Academic Editor

PLOS One